# Unsupervised Deformable Image Registration Revisited: Enhancing Performance with Registration-Specific Designs

**Hengjie Liu**[1,2,3] (iD)                                       HENGJIELIU@MEDNET.UCLA.EDU

[1] *Physics and Biology in Medicine, University of California, Los Angeles*

[2] *Department of Radiation Oncology, University of California, Los Angeles*

[3] *Department of Radiation Oncology, University of California, San Francisco*

**Dan Ruan**[1,2]                                                DRUAN@MEDNET.UCLA.EDU

**Ke Sheng**[3]                                                  KE.SHENG@UCSF.EDU

**Editors:** Accepted for publication at MIDL 2025

## Abstract

Deformable image registration (DIR) is ill-posed. Many registration-specific designs and regularizations, whose rationale carries across classic optimization methods to deep-learning-based (DL) frameworks, are crucial to registration performance. This paper presents a comprehensive "ablation" type study to pinpoint the key drivers for unsupervised monomodal DL-DIR. We conducted controlled experiments and benchmarked performance against state-of-the-art methods. Our findings highlight the benefits of multi-resolution pyramids, local correlation, and inverse-consistency constraints, and demonstrate that simple network architectures can achieve strong performance—even with far less training data. The code will be publicly available at: Unsupervised-DL-DIR-Revisited.

**Keywords:** Deformable Image Registration, Unsupervised Learning, Benchmark, Ablation Study

## 1. Introduction

Deformable image registration (DIR) involves spatially aligning images using non-linear deformation fields and is critical in many medical image analysis tasks. Deep-learning-based DIR (DL-DIR) has recently risen in popularity, with unsupervised and weakly supervised training becoming the dominant approaches. Numerous novel network architectures have been proposed (Mok and Chung, 2020; Kang et al., 2022; Chen et al., 2022; Jia et al., 2022; Shi et al., 2022; Guo et al., 2024; Tan et al., 2024). However, Jena et al. (2024) shows that well-optimized conventional methods perform on par with or surpass many unsupervised DL-DIR methods. In addition, Jian et al. (2024) demonstrates that high-level registration-specific designs such as multi-resolution pyramid and correlation calculation are more critical than the choice of low-level computational blocks (e.g., U-Net vs. Transformer vs. Mamba). Notably, the latter study was restricted to a weakly supervised setting, where label-matching supervision could skew results in favor of less competitive architectures. These observations raise key questions about whether unsupervised DL-DIR can truly surpass conventional methods—and, if so, which components drive that performance.

To elucidate these questions, we conduct a comprehensive evaluation of different deformation estimation blocks under controlled conditions. Our findings reveal that simple architectures, when combined with effective registration-specific design elements, can deliver state-of-the-art (SOTA) performance, even with far less training data. This suggests that future efforts should prioritize refining registration-specific strategies over pursuing increasingly complex architectures.

## 2. Methods and Datasets

We employ three registration-specific designs that have proved their efficacy: the multi-resolution pyramid, correlation calculation, and inverse consistency setup. The multi-resolution coarse-to-fine pyramid strategy from conventional methods is now commonly used in DL-DIR, where one standard practice is to employ a dual-stream feature encoder for image pairs and apply progressively upsampling and warping for deformation prediction (Kang et al., 2022; Honkamaa and Marttinen, 2024; Tan et al., 2024). We refer to this dual-stream pyramid architecture as DP. Local feature correlation inspired by optical flow studies has shown promise in improving registration accuracy (Kang et al., 2022; Jian et al., 2024). Liu et al. (2024) further introduced vector field attention (VFA), a novel deformation estimation paradigm that directly computes displacement vectors using correlation as weights. Finally, symmetry and inverse consistency constraints have demonstrated superior performance in both conventional methods (Avants et al., 2008) and DL-DIR (Honkamaa and Marttinen, 2024), as they provide a robust inductive bias and regularization effect that promotes smoother, more realistic deformations.

Following these guidelines, we experimented with different deformation prediction modules in a controlled setting, as shown in Figure A1. We universally adopted the DP setting and used the same standard residual U-Net encoder for multilevel feature extraction. For deformation prediction, we experimented with three settings: (a) residual convolution blocks, (b) residual convolution blocks with built-in inverse consistency, and (c) VFA. The built-in inverse consistency in (b) follows SITReg (Honkamaa and Marttinen, 2024) while uses the same network as (a). For inputs of (a) and (b), we can concatenate moving and fixed features (denoted as MF), further concatenate correlation (MFC) or use correlation only (C). In total, we experimented with 6 configurations, as shown in Table 1.

Experiments were conducted on two publicly available brain MR datasets from the Learn2Reg challenge: OASIS and LUMIR. The 414 images of OASIS were split into 300/30/84 for training, validation, and testing. For testing, we randomly sampled 200 pairs from the 84 images. For LUMIR, the training includes 3,384 images and and we submit to challenge leaderboard to evaluate on the validation set (38 pairs). For comparison, we adopted two widely used methods, VoxelMorph (Balakrishnan et al., 2019) and TransMorph (Chen et al., 2022), as well as two SOTA methods from the Learn2Reg LUMIR challenge: the best baseline VFA (Liu et al., 2024) and the winning method SITReg (Honkamaa and Marttinen, 2024). In addition, we included a conventional baseline called Greedy (Yushkevich et al., 2016), which was the best conventional method reported in Jena et al. (2024).

## 3. Results

Results on the OASIS dataset (Table 1 and Figure B3) show that all proposed variants achieve competitive performance, matching SOTA accuracies while outperforming the conventional Greedy method and baselines lacking registration-specific designs (VoxelMorph and TransMorph). The correlation-only models (DP-Conv-C and DP-ConvIC-C) surpasses their counterparts with more parameters, highlighting the role of correlation in deformation estimation. This also explains the success of the DP-VFA variant, which directly extracts deformation from correlation via VFA with substantially fewer trainable parameters. The Appendix includes LUMIR results and further experiments on model scaling and training

set size. Notably, competitive performance can be attained with substantially less training data when registration-specific designs are properly applied.

Table 1: Registration results for the OASIS brain MRI dataset (200 test pairs). The proposed variants list parameter counts in the "(encoder/decoder) total" format. Metrics in **bold** denote the best-performing methods, while those underlined are competitively close.

|  | Dice ↑ | HD95 ↓ | SDlogJ (×100) ↓ | NDV (%) ↓ | Params (M) |
|---|---|---|---|---|---|
| Initial | 0.5759 (0.0682) | 3.95 (0.95) | - | - | - |
| Greedy | 0.8068 (0.0297) | 2.02 (0.56) | **13.18 (0.95)** | **0.0007** | - |
| VoxelMorph | 0.7647 (0.0392) | 2.55 (0.72) | 21.96 (2.77) | 1.27 | 0.30 |
| TransMorph | 0.7934 (0.0276) | 2.15 (0.56) | 17.00 (1.79) | 0.83 | 46.56 |
| VFA | 0.8203 (0.0233) | 1.87 (0.45) | 14.00 (0.89) | 0.065 | 2.01 |
| SITReg (IC) | 0.8230 (0.0232) | 1.81 (0.45) | **12.98 (1.00)** | 0.027 | 15.08 |
| (a)DP-Conv-MF | 0.8237 (0.0237) | 1.82 (0.46) | 15.32 (1.02) | 0.33 | (0.51/2.35) 2.85 |
| (a)DP-Conv-MFC | **0.8281 (0.0227)** | **1.79 (0.45)** | 15.63 (1.12) | 0.37 | (0.51/2.66) 3.17 |
| (a)DP-Conv-C | **0.8283 (0.0226)** | **1.79 (0.46)** | 14.64 (0.98) | 0.33 | (0.51/1.49) 1.99 |
| (b)DP-ConvIC-MF | 0.8223 (0.0244) | 1.82 (0.47) | **12.98 (0.98)** | 0.027 | (0.51/2.35) 2.85 |
| (b)DP-ConvIC-C | 0.8244 (0.0225) | 1.80 (0.45) | **12.79 (1.01)** | 0.028 | (0.51/1.80) 2.31 |
| (c)DP-VFA | 0.8199 (0.0237) | 1.87 (0.46) | 13.91 (0.94) | 0.031 | (0.51/0.28) 0.79 |

## 4. Discussion and Conclusion

Registration-specific designs—such as multi-resolution pyramids, correlation computation, and inverse consistency constraints—are essential for achieving robust performance in unsupervised monomodal DL-DIR. Notably, multi-resolution refinement and inverse consistency constraints serve as effective regularizers for the inherently ill-posed DIR and should be incorporated whenever possible. Additionally, our results show that models leveraging only correlation-based features (e.g., DP-Conv-C, DP-ConvIC-C, DP-VFA) are particularly promising. This suggests that commonly used methods, which directly apply convolutions to fixed and moving image features, may expend unnecessary capacity to address model mismatches in displacement prediction, whereas exploiting feature correlations directly offers a more efficient and targeted solution.

Our study also points to promising avenues for future work—particularly the development of novel registration-specific strategies and their integration into a cohesive, synergistic framework. Further integrating deep-learning-based representation learning with conventional optimization schemes is also promising and has already produced encouraging results in recent studies (Jena et al., 2025; Siebert et al., 2025; Xin et al., 2024).

## Acknowledgments

The current project is funded by NIH R01CA188300 and DOD W81XWH2210044.

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

## Appendix A. Extended Methods

**Architectural Details** Figure A1 illustrates the underline{d}ual-stream underline{p}yramid (DP) architecture incorporating the same feature encoder with various deformation prediction blocks. We fix the number of levels to be 5 (i.e., 4 levels of downsampling are used). We intentionally avoid the use of large convolution kernels or self/cross-attention mechanisms in the feature encoder and deformation decoder and keep them as simple as possible. The encoder is a standard U-Net encoder with residual connections. It employs standard $3\times3\times3$ convolutions with stride-2 downsampling, along with InstanceNorm and LeakyReLU (with a negative slope of 0.2). For the deformation decoder, types (a) and (b) use two layers of standard $3\times3\times3$ convolutions with residual connections, followed by a final layer that outputs the deformation field. Specifically, the deformation output layer is implemented with a kernel size of 3, stride of 1, 3 output channels, weights initialized from a normal distribution (mean 0, variance 1e-5), and biases set to zero. The inputs for types (a) and (b) may consist of concatenated fixed and moving features (MF), optionally augmented with correlation (MFC) or correlation only (C). In the inverse consistent setup (b), correlations are computed

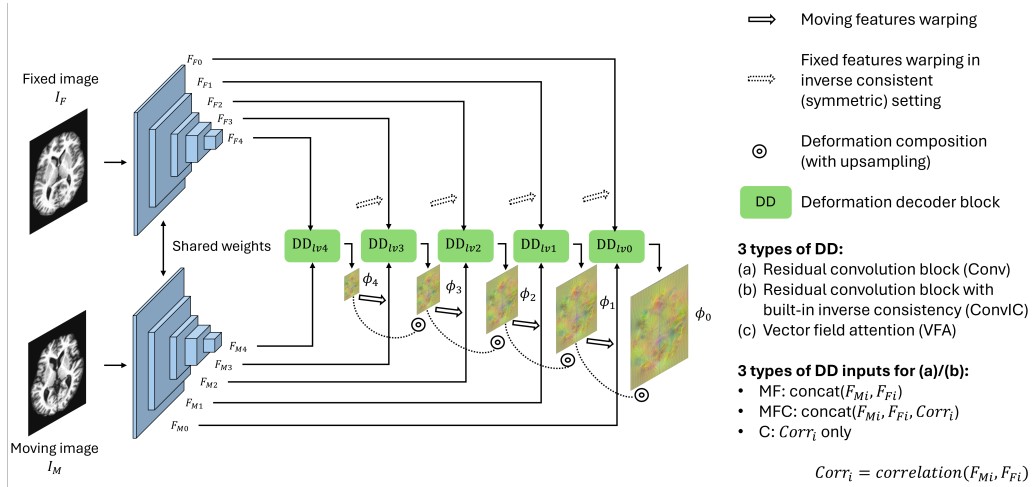

Figure A1: The dual-stream pyramid (DP) architecture with the same feature encoder and different variations of deformation decoders.

in both directions. Due to memory constraints, the DP-ConvIC-MFC variant is not tested. For type (c) DP-VFA, the moving and fixed features are first processed through a $3 \times 3 \times 3$ convolutional projection layer before being fed into the deterministic VFA module.

**Training Details** All experiments were conducted on NVIDIA RTX 6000 Ada GPUs. In each epoch, 100 fixed/moving image pairs (50 pairs evaluated in both directions) were randomly sampled for model training. We employed the same loss function, $Loss = L_{sim} + \lambda * L_{reg}$, with $1 - NCC$ as similarity loss and diffusion as smoothness regularization. $\lambda$ was empirically set to 1. The random seed of training was set to 42 to ensure consistent training data across all experiments. For OASIS, all models were trained for 200 epochs using a constant learning rate of $10^{-4}$, and the model achieving the best validation Dice score was selected for final testing. For LUMIR, all models were trained for 1,500 epochs using cosine annealing with warm starts (learning rate between $10^{-3}$ and $10^{-5}$), and the model with the lowest validation $L_{sim}$ was saved.

## Appendix B. Extended Results

**Effect of Model Scaling** Figure B2 (left) demonstrates the scaling effect for VoxelMorph and the proposed variant DP-Conv-MF. Both models benefit from increased model size; however, DP-Conv-MF exhibits a significantly higher baseline performance. TransMorph is used as the reference.

**Effect of Training Set Size** There is currently no consensus on the ideal dataset size for training unsupervised DL-DIR models, with most studies using as much data as possible. For inter-subject registration, $N * (N - 1)$ image pairs can be generated with $N$ images (consider both directions), so we can potentially train registration networks with a small dataset. Using our competitive benchmark DP-Conv-MF, we assessed the impact

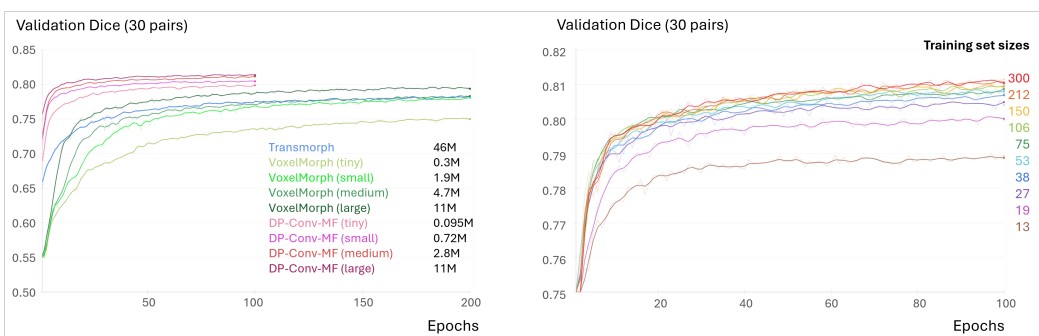

Figure B2: Left: the effect of scaling up model size (parameter count) for VoxelMorph and proposed variant DP-Conv-MF. The training set size is fixed to 300. Right: the effect of training set size. The model is fixed to be DP-Conv-MF (medium).

of training set size of OASIS by reducing it in steps proportional to a factor of $\sqrt{2}$, with each smaller set being a strict subset of the larger one. As shown in Figure B2 (right), the performance gains diminish as the training sample size exceeds 50. We performed a similar experiment on LUMIR using the DP-ConvIC-C variant and the quantitative results are summarized in Table B3. Notably, in both cases, the proposed variants trained on only $\sim 10$ images achieved comparable Dice with TransMorph trained on the entire dataset. This result suggests that with a well-designed architecture, competitive performance can be achieved with much less training data.

Table B2: Registration results for the LUMIR brain MRI dataset (38 validation pairs). Superscripts indicate sources: official baselines[1], challenge winners[2], and our re-trained/reported baselines and our proposed variants[3]. SITReg-v1 is the vanilla version trained with NCC and diffusion loss. SITReg-v2 is the final challenge-winning version, further trained with group consistency and NDV loss (not used by other methods in this table).

| Model | Dice ↑ | HD95 ↓ | TRE (mm) ↓ | NDV (%) ↓ |
|---|---|---|---|---|
| **Official baselines** | | | | |
| VoxelMorph[1] | $0.7186 \pm 0.0340$ | 3.9821 | 3.1545 | 1.1836 |
| TransMorph[1] | $0.7594 \pm 0.0319$ | 3.5074 | 2.4225 | 0.3509 |
| VFA[1] | $0.7726 \pm 0.0286$ | 3.2127 | 2.4949 | 0.0788 |
| **Challenge-winning** | | | | |
| SITReg-v1[2] | $0.7742 \pm 0.0291$ | 3.3039 | 2.3112 | 0.0231 |
| SITReg-v2[2] | $0.7805 \pm 0.0287$ | 3.1187 | 2.3005 | 0.0025 |
| **Our re-trained/reported baselines[3]** | | | | |
| Greedy | $0.7531 \pm 0.0334$ | 3.6953 | 2.2994 | 0.0004 |
| VFA | $0.7734 \pm 0.0286$ | 3.2063 | 2.4739 | 0.1051 |
| SITReg-v1 | $0.7727 \pm 0.0284$ | 3.3319 | 2.3120 | 0.0308 |
| **Our variants[3]** | | | | |
| (a) DP-Conv-MF | $0.7713 \pm 0.0290$ | 3.3534 | 2.4676 | 0.4158 |
| (a) DP-Conv-MFC | $0.7730 \pm 0.0291$ | 3.3566 | 2.4449 | 0.4672 |
| (a) DP-Conv-C | $0.7747 \pm 0.0295$ | 3.3666 | 2.4135 | 0.3795 |
| (b) DP-ConvIC-MF | $0.7717 \pm 0.0288$ | 3.3489 | 2.3660 | 0.0310 |
| (b) DP-ConvIC-C | $0.7724 \pm 0.0288$ | 3.3873 | 2.3357 | 0.0309 |
| (c) DP-VFA | $0.7764 \pm 0.0284$ | 3.2157 | 2.4420 | 0.0540 |

Table B3: Registration performance on the LUMIR brain MRI validation set (38 pairs) using varying training set sizes. The (b) DP-ConvIC-C variant is used for all experiments. **Bold** indicates the best value in each column.

| Training set size | Dice ↑ | HD95 ↓ | TRE (mm) ↓ | NDV (%) ↓ |
|---|---|---|---|---|
| all data / 3,384 | $0.7724 \pm 0.0288$ | 3.3873 | 2.3357 | 0.0309 |
| 1,000 | $\mathbf{0.7726 \pm 0.0283}$ | 3.3772 | **2.3088** | 0.0332 |
| 300 | $0.7725 \pm 0.0287$ | **3.3756** | 2.3208 | **0.0303** |
| 100 | $0.7717 \pm 0.0280$ | 3.3966 | 2.3305 | 0.0319 |
| 50 | $0.7705 \pm 0.0286$ | 3.4162 | 2.3479 | 0.0304 |
| 30 | $0.7690 \pm 0.0287$ | 3.4576 | 2.3723 | 0.0328 |
| 20 | $0.7674 \pm 0.0288$ | 3.4521 | 2.4053 | 0.0358 |
| 10 | $0.7642 \pm 0.0308$ | 3.5216 | 2.3771 | 0.0361 |

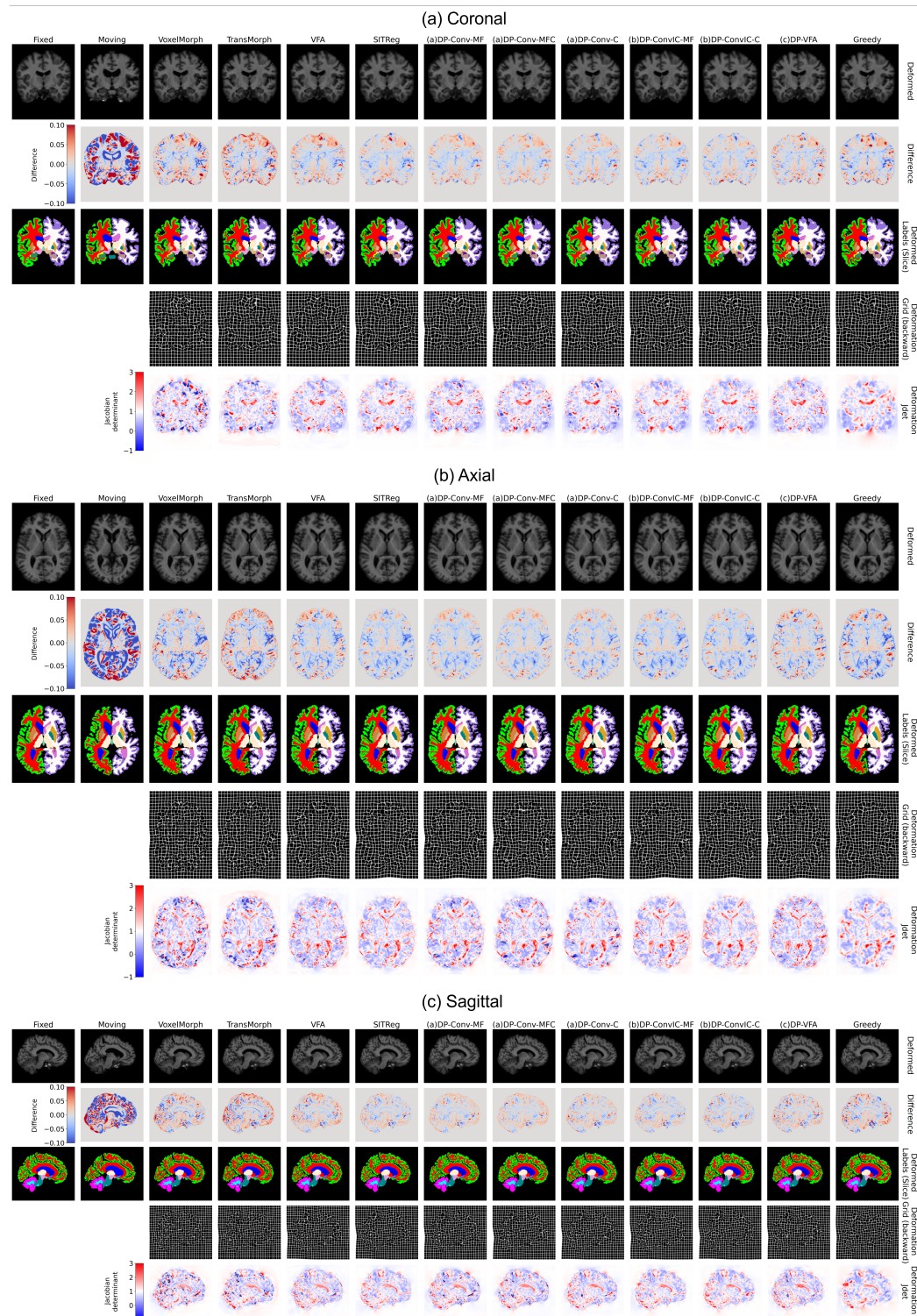

Figure B3: Qualitative results of a test case in OASIS. The last row shows the Jacobian determinant maps with the non-positive regions contoured in black.

