# OpenReview forum: "Unsupervised Deformable Image Registration Revisited: Enhancing Performance with Registration-Specific Designs"
_MIDL.io/2025/Short_Papers — MIDL 2025 - Short Papers_

### Official Review · Reviewer_jL39 · 2025-04-28

**Rating:** 5
**Confidence:** 5

**Summary:**

This paper presents a study revisiting unsupervised deformable image registration and investigating important roles of registration-specific design strategies, such as multi-resolution pyramids, correlation computation, and inverse consistency constraints. The experimental results show that simple architectures, when properly designed, can achieve state-of-the-art results when compared with selected baselines.

**Strengths:**

- The paper is well written and organized, and provides valuable empirical results that could benefit the MIDL community.

- The focus on registration-specific designs rather than unnecessary model complexity is important to explore for ill-posed registration methods.

**Weaknesses:**

While the topic investigated in this paper is valuable to the MIDL community, the experimental results would be more compelling if validated on a broader and more diverse set of large-scale datasets. Additionally, the range of baseline comparisons is somewhat limited, likely due to the space constraints of the short paper format. Overall, while the findings are promising, they would be significantly strengthened by more comprehensive experimental evaluations in future work.

---

### Decision · Program_Chairs · 2025-05-01

Accept